# Gamification and Hazard Communication in Virtual Reality: A Qualitative Study

**DOI:** 10.3390/s21144663

**Published:** 2021-07-07

**Authors:** Janaina Cavalcanti, Victor Valls, Manuel Contero, David Fonseca

**Affiliations:** 1Institute for Research and Innovation in Bioengineering (I3B), Universitat Politècnica de València, Camino de Vera s/n, 46022 Valencia, Spain; cjanaina@gmail.com (J.C.); mcontero@upv.es (M.C.); 2Architecture Department, La Salle Campus Barcelona, Ramon Llull University, 2 Quatre Camins St., 08022 Barcelona, Spain; victor.valls@salle.url.edu

**Keywords:** immersive player experiences, serious games, human-computer interaction, usability, user experience, warnings, learning engagement, head-mounted display, gamification methodologies, educational games

## Abstract

An effective warning attracts attention, elicits knowledge, and enables compliance behavior. Game mechanics, which are directly linked to human desires, stand out as training, evaluation, and improvement tools. Immersive virtual reality (VR) facilitates training without risk to participants, evaluates the impact of an incorrect action/decision, and creates a smart training environment. The present study analyzes the user experience in a gamified virtual environment of risks using the HTC Vive head-mounted display. The game was developed in the Unreal game engine and consisted of a walk-through maze composed of evident dangers and different signaling variables while user action data were recorded. To demonstrate which aspects provide better interaction, experience, perception and memory, three different warning configurations (dynamic, static and smart) and two different levels of danger (low and high) were presented. To properly assess the impact of the experience, we conducted a survey about personality and knowledge before and after using the game. We proceeded with the qualitative approach by using questions in a bipolar laddering assessment that was compared with the recorded data during the game. The findings indicate that when users are engaged in VR, they tend to test the consequences of their actions rather than maintaining safety. The results also reveal that textual signal variables are not accessed when users are faced with the stress factor of time. Progress is needed in implementing new technologies for warnings and advance notifications to improve the evaluation of human behavior in virtual environments of high-risk surroundings.

## 1. Introduction

Warnings are tools that are frequently used to convey information about hazards when they cannot be designed out or guarded against [1]. Therefore, warnings need to influence people to act in a way to avoid personal injury and property damage. For this reason, their use is more necessary in a complex, hostile environment with communication blocks or interference and/or a stressful environment [2]. Effective warnings should rapidly attract attention, elicit knowledge, and enable compliance behavior (i.e., lead to appropriate decisions regarding performance execution). Previous studies have found that familiarity, stress, time pressure, or the presence of other mental activities can interfere with and reduce warning compliance rates [3,4]. Nevertheless, warning systems usually comprise passive elements that are always visible or, sometimes, dynamic lighting [2]. Advances in technology have produced a range of tools that can be used in this area to increase performance, evaluate user experience, learning and training, and control sensors and smart systems [5,6,7,8].

New technologies have also brought advances in the field of virtual reality (VR) by rapidly breaking spatial-temporal barriers and expanding its use and scope. The costs of VR consistently decrease as a consequence of the development of low-cost headsets [9], for example. The use of mobile technology to stimulate emotion, training, education, and health and to evaluate decision-making [10,11,12,13,14,15] has been increasing rapidly. Its use is especially highlighted when the physical presence offers some kind [16] of risk or extra cost. Environmental simulations make it possible to study these scenarios under controlled laboratory conditions [17]. Additionally, VR has become an important tool for participatory design, facilitating the evaluation of a product and the involvement of users in different development stages [18,19,20]. We believe that, when used with a qualitative methodology, VR becomes particularly relevant because it allows individuals to reflect and to expose their opinions. It is therefore possible to obtain detailed information with high reliability about the product or technology [21], making it easy to understand user behavior.

Researchers have recently paid attention to the playability and appeal of gamification techniques for human behavior assessments [22,23,24]. Coming from the video games and games, but differing of them by their non-entertainment purpose [24], these techniques are also becoming an important tool for training because they can be closely aligned with the design of good educational experiences [25,26]. In addition, they allow players to naturally produce rich sequences of actions while performing complex tasks by drawing on their competencies [27]. Gamification techniques are interactive [28], which is a key for motivation. Indeed, it is possible to say that game mechanics are linked to human desires, challenges, rewards, status, competition, success, and self-expression.

On the other hand, the user approach to user research techniques is mainly focused on the study of behavioral goals in work settings. For this reason, the task becomes the pivotal point of user-centered analysis and evaluation techniques [29]. The term user experience (UX), popularized by Don Norman [30], includes the feelings and meaningful aspects of user interaction with machines and services. Qualitative techniques enable the assessment of users’ degree of satisfaction, motivation, and interest with the item investigated. Currently, UX methods do not necessarily include the participation of the end user in the creative process of the product. We believe it is essential to define the causes of individuals’ behaviors and decisions in tasks. The Socratic model of postmodern psychology is valuable because it relates to data and addresses complex information about the product, experience, or technology studied [31]. The user is asked to reflect on a specific topic through a dialog between the user and the interviewer.

Originating from a constructivist paradigm, the qualitative method of bipolar laddering (BLA) acquires information from users themselves rather than only from observation of their behavior. Defined as a psychological exploration technique, BLA allows for the identification of key experience factors and how they are linked in a person’s mind [32]. Thus, BLA aims to identify the critical factors of any user experience. It operates using open-ended answer classifiers with positive/negative poles to define the strengths and weakness of systems [33]. This makes participative product design possible and promotes user participation in the testing of product design. Users and facilitators work together to define important aspects through strategic conversations. In this work, we adopted this technique to uncover how the qualities of the system, the implications of use, and personal values are connected in the minds of individuals [31].

In this paper, we present an immersive game (Game for Safety) that consists of a virtual environment endowed with signage and risks in which it is possible to extract user information and improve learning. This work born as an adaptation of the application created for the Game4City 3.0 project, which is being developed (2016–2021) by the Barcelona School of Architecture (Catalonia Polytechnic University, ETSAB-UPC) and the School of Architecture of La Salle—Ramon Llull University (ETSALS-URL) [34,35,36]. Game4City applies VR strategies to design 3D indoor and outdoor spaces for educational purposes [31,36,37,38,39,40]. Its aim is to show how the implementation of “gamified” virtual strategies in architecture, facilities, construction, and urban design can increase students’ spatial comprehension as well as other characters as citizens or professionals, increasing their interest in the collaborative/gamified/interactive design of spaces.

Inspired on Game4City, Game for Safety implements sensors in signaling systems with the aim of increasing obedience, perception and, consequently, efficiency. Therefore, this paper aimed to establish a UX assessment and answer the following research questions:RQ1: Are there differences in the perception, behavior and usability in users with regard to different variable signs as a function of the level of danger?RQ2: What elements could improve the design of a virtual environment for the analysis of user behavior in a gamified risk context?RQ3: Which components of an immersive serious risk game can impact the user?

For this purpose, we propose some new configurations and variables for warnings that make them dynamic and smart to improve human interaction. By making use of a serious game in which the user interacts with the proposed signaling system and distinct dangers, we evaluate signals and game usability. Users execute a VR task while their performance is recorded using their navigation data and later exported into a video. This video is stored for further analysis. Using BLA methods, we explore users’ deep reflections [41], identify perceptions, and compare them with behavioral data to evaluate the user experience [42,43].

This work is organized as follows. Section 2 describes related work. In Section 3, we present the methods, methodology and evaluation procedures. The results of the experiments are presented in Section 4. Section 5 presents our conclusions and future work.

## 2. Related Work

Several studies have reported the success of games for learning and skill development [26,44,45,46]. The fact that a game environment is full of stimuli makes it an excellent tool to improve learning procedures. Games are flexible and can be consumed in a wide range of locations (e.g., home, school), and they present the opportunity to learn by experimenting and exploring [44]. This is in line with the contemporary context that emphasizes important results during the education process [47,48,49]. Based on this, VR games could provide a comprehensive training environment that brings new opportunities to safety teaching and learning processes [50].

Previous studies have shown that immersive serious games can promote better knowledge retention with the aim of aviation safety [51]. In this study, an immersive serious game is compared with a traditional education method. The serious game presents a better result for learning both immediately after the experience and for a longer period of time afterward. Greater engagement and fear arousal are identified as factors that contribute to explaining the findings.

In a previous study, a 3D game environment for construction engineering safety education greatly improved future construction personnel’s safety competency [52]. The refereed study was divided into three modules. In the first one, named safety knowledge dissemination, the students acquired the knowledge. The second task consisted of reflecting on the knowledge acquired during the first module and identifying the risks in an interactive environment (safety knowledge reflection). Finally, a knowledge assessment was conducted that required participants to resolve the hazards. The technology employed was not immersive.

A non-immersive virtual environment was also the object of study about risk assessment among university students in a chemical plant [53]. The results showed that users of the virtual environment were able to identify more risks in a real environment than nonusers.

Immersive virtual reality is an adequate and important tool that facilitates the points mentioned previously. The use of this type of mechanism for safety training prevents misunderstanding in real situations [54]. A study of emergency exits verified the effect of cause-consequence actions. By training in a virtual environment, users already knew the better action to take in a real hazard environment. However, for the success of a gamified proposal, it is necessary to use a range of variables, such as technological requirements and the design itself [46,55,56], which depends on the dynamic to be gamified, the context or device to be used, and the user´s profile.

In general, static signs made of paper, metal or plastic [57] are the main support system for safety communications. These are symbol-based signs consistent with the International Organization for Standardization (ISO) 3864-1 and American National Standards Institute (ANSI) Z535. Some studies have examined the effect of dynamic features in signs on behavioral compliance during a work-related task and emergency [3,58,59,60]. The findings suggested that dynamic presentations produced higher compliance than static presentations [58,61,62,63,64,65] mainly because of some features that make them more noticeable. This is in line with attention theory [66].

Other new alternatives are smart active systems or smart signals, concepts that have emerged from smart cities [8]. A smart signal is usually a traditional signal to which some smart capabilities are added, enabling an optimal match between occupants´ behavior and characteristics and environmental conditions [2,57]. The smart signal can be dynamic, adaptable and/or interactive. In a dangerous situation, this could be especially useful because people tend to narrow their attention in stressful situations. Salience could make smart signs more noticeable and more resistant to habituation (availability only when necessary, for instance). In the scope of this work, we focus on smart warnings in building sites.

Gamified environments have previously been used as a tool for training risk management. They have been used, for example, to help students interpret their activity, both in real time and later, through cause-consequence analysis [67,68,69]. Additionally, some VR environments have been used to evaluate products [19,55,70]. However, the efficacy of warning signals in a simulated environment has rarely been explored in research [60,71]. Additionally, very few works have used a gamified VR environment with a qualitative assessment to understand and evaluate usability and cause-consequence during the assessment.

## 3. Materials and Methods

From a methodological point of view, we adopted a user-centered design. The user´s experience was appraised with a qualitative approach. The objective was to evaluate behavior and define its causes [31]. This study privileged measures of behavior (attention, performance, errors and learnability) and satisfaction (qualitative assessment and subjective responses after game interaction), as summarized in Table 1.

The experiment consisted of testing the VR environment. Users crossed an office where an accident had just occurred, acting around the risks (run away, face or solve) and warnings (perceiving, reading and obey). Their explicit task was to cross the maze in the shortest time. They were informed that they could solve some risks (e.g., extinguish a fire), but this was not a mandatory action.

Prior to the game experience stage, demographic data and characteristics were collected with the aim of discarding individuals who, in some way, could compromise the research and identifying characteristics that could jeopardize the results. Additionally, data on behavioral traits, such as behavioral intention and compliance, were collected. This information clarified when signaling variables could improve users’ behavior and when the behavioral increase is the result of a propensity of individual behavior.

Once the users left the immersive office, a retention task was assessed in which the participants had to name the signals presented at the virtual office. More details will be explained next.

### 3.1. Components

Regarding the components required for this proposal, we took into account two premises:Implementing a device system that allows easy use anywhere and is easily recognized, familiar and affordable;Technological devices, tracking and mobile systems with a high degree of accuracy in stimuli reproduction to create a sense of presence [72,73].

Based on this, we chose sensor-type HDMs for VR headsets. We used the HTC Vive headset because it is designed to track movement freely with high precision through a space of 4 m × 4 m [74]. It consists of a headset with two controllers and two base stations that emit infrared pulses at 60 Hz, providing submillimeter tracking precision to the headset and controllers [30]. The headset has the following characteristics: two OLED panels, each with a display resolution of 1080 × 1200 pixels per eye update at 90 Hz cover a send out horizontal and vertical infrared laser sweeps spanning 120° in each direction, a mass of 470 g, G-sensor, gyroscope, proximity, remote control with a battery with a range of 6 h of play, and SteamVR tracking sensors [75]. The differences in time at which the laser hits the various photodiodes allows recovery of the position and orientation of headset.

HTC Vive has development compatibility with the Unity and Unreal Engine 4 game engines. For this work, we chose Unreal because of its more straightforward coding method (C++ coding allows the possibility of coding through blueprints and Visual Studio) [30,76,77]. Unreal is developed by Epic Games and is free for use for academic purposes. Figure 1 shows the components selected to implement the experiment.

The character movement adopted is lineal character movement. We choose this as it is the simplest method and reduces VR sickness. It consists of making the character move forward immediately when the user pushes the forward button [30].

### 3.2. Definition of the Participants’ Profile

To define the sample size of this study, it was necessary to take into account the saturation principle [65], which defines the extent to which the data collection provides new information in terms of the setting and sample. The purpose of the research was to obtain an in-depth understanding of users’ perceptions and to obtain better precision in the description. A large sample size may lead to a repetition of information [56]. On the other hand, users’ actions have fundamental relevance for the study. Therefore, special attention was paid to selecting an expert, active sample of professionals with a recognized reputation.

Considering this, the sample was composed of 10 (5 male and 5 female) volunteer specialists in risk management/architecture/engineering with more than 5 years of experience. All of them agreed to collaborate with the research and approved the informed consent to conduct the evaluation. Twenty percent of the sample had never used an HMD, while 80% had tried it once. All participants reported normal or corrected-to-normal vision and ranged in age from 31 to 65, as demonstrated in Figure 2. They all reported that they had no physical or mental conditions that they believed would prevent them from participating in a VR immersive game simulation.

With regard to game experience and technological knowledge, 20% of the users were frequent players. 3D technologies were frequently used by the participants, not for game purposes but for professional reasons, as shown in Figure 3.

### 3.3. Virtual Environment Game for Safety

To increase the involvement of the subjects, a game-based virtual environment was designed by a multi-disciplinary team composed of designers, programmers, multimedia animators, and specialists in educational technology and human computer interaction (HCI). We aimed to create an atmosphere of fear, because emotional arousal (especially with negative emotions) positively affects presence [78,79,80] and retention [79,81]. However, we aimed to prevent discomfort for the users to avoid diminished performance on difficult tasks [82]. Special attention was paid to the context, design, navigation, task to be performed, presence, involvement of the subject, and preventing sickness.

Based on this, we opted to create one environment composed of just one entrance and one exit, but a range of different halls to choose. The halls are similar, but the user does not know this. This helps to increase the user disorientation without produce physical discomforts. In addition, it allows evaluation of the behaviors of the player after receiving different types of information.

The application is based on a framework, which involves the following game elements [83]:Dynamics—a narrative context communicated by both prior oral orientation and posters on the pre-experimental training room.Mechanics—the challenge of cross the maze in the shortest which drive the action forward and generate engagement.Components—the specific instantiations of mechanics and dynamics. In Game for Safety, particle effects were used to show the extra time obtained by the user during the game.

The setting, a maze, is reminiscent of a workplace, with walls similar to office partitions and a vinyl floor (purple, gray and black, with a different color for each section) with office elements (photocopier, coffee machine and water). The users were free to act based on their decisions. The character movement was linear; when the user pushed the forward button, the character immediately moved forward.

The first action in the environment was a pre-experimental training section. It consisted of a navigation task designed to eliminate any stress and anxiety and to introduce and familiarize the user with the technology adopted. This task was performed in a room composed of a main hall with a similar layout, as shown in Figure 4, that was specifically designed for training purposes. Posters were displayed that reported an incident that occurred in the office as well the necessary commands for navigation and the task.

After testing the commands, the subject immediately entered the principal maze, which was composed of three areas with the same configurations and same hazards but distinct variable signs. The signals were displayed near a referenced hazard (for which the user was alerted about the risk) or at strategic points (educative and route signals). Figure 5 shows the disposition of hazards, routes and educative signals. To guarantee that the subjects went through all the signals (experimental conditions), the labyrinth was mirrored (if the user chose right or left, he or she went through the same risk and the same warning). A command was included that added walls to prevent the user from returning to the same section. The walls appeared discreetly and were imperceptible to the user.

The VR Game for Safety allowed individuals to execute the tasks while security signals and hazards were presented, as shown in Figure 6. The warning design variables respected the following characteristics and were placed in such a way that each subject could visualize all signal types:Static Sign—respect ANSI and ISO rules;Dynamic Sign—the ISO sign with intermittent red-, yellow- and white-LED lights;Smart signals—signal available only after crossing a certain point.

The experimental conditions were created with reference to conventional ANSI Z535.2 and ISO-type warnings of a wet floor, fire, height, fire classification and indicative arrow. The selected warnings and a description of their variables can be seen in Table 2. They were displayed near the referenced hazard (for which the user was alerted about the risk) or at strategic points (educative and route signals).

The participants were asked to cross the maze in the shortest time possible. The time was used to increase the challenge aspect. They were verbally warned that there were hazards on the way. Specifically, with regard to the fire risks, they were informed that they would be able to solve the issue using the fire extinguisher (and earn a bonus of an extra 10 s) or just pass by. The player knows the amount of bonus time gained through an explosion of a green “+10”. For this we used Cascade, the participle effect creation software from Unreal 4. If they assumed a risky position, they lost 10 s.

To evaluate both the virtual environment and the variables of the signals, we proceeded to evaluate the user-product relationship. In this particular case, the sample selected was the user and the virtual environment, and the warnings were the products/services. The users´ desires, needs and goals were assessed by techniques for obtaining and systematizing data to assess the users’ experience and identify necessary improvements [29].

### 3.4. The Experiment Protocol

The experiment was conducted over 5 days (two users per day) at the end of the morning and at the end of the afternoon to maintain maximum similarity of the conditions. The experimental session was divided into four phases: the introduction, pre-experimental training session, experimental session, and follow-up interview, as follows:

#### 3.4.1. Stage 1 Introduction (5 min)

Participants gave informed consent in an anonymous form and then completed a questionnaire about demographics, behavioral intentions (safety behavior, safety experience, work situational assessment), game experience, and technological knowledge. The participants were told that the main objective of the study was to evaluate the warnings and the virtual environment, main tasks, and game mechanism.

#### 3.4.2. Stage 2 Pre-Experimental Training Session (3 min)

For safety reasons arising from the current situation of COVID-19, we proceeded to sanitize and disinfect the devices. Then, the participants were placed in the pre-salon and asked to explore this room freely until they felt they were able to control the input device. When the exploration ended, the recording of participant´s navigation data was launched.

#### 3.4.3. Stage 3 Experimental Session (10 min)

Stage 3 began after the participants reported that they were comfortable with the devices and able to begin. They crossed the START point, and the counting activation began. The users were free to verbalize their actions in this phase and we carefully took notes for posterior analyses.

#### 3.4.4. Stage 4 Follow-up Interview (10 min)

The participants were asked about their first impression and then began the BLA. With this method, we aimed to determine how the product, the consequences of its use, and the personal assessment of the product were related to the user´s thinking [84]. For the BLA technique, we followed three steps:Collection: This step consisted of a blank template for the positive elements (strengths) and another identical template for the negative elements (weaknesses). The interviewer asked the participants to state which aspects of the experiment and signals they liked best or that helped them in their tasks and what elements they disliked or that disturbed the task. We limited the elements to five positives and five negatives for each person. The elements were summarized in one word or a short sentence.Assessment: Once the list of positive and negative aspects was finished, the interviewer asked the user to rate each aspect on a scale using a score between 0 (lowest) and 10 (most).Definition: At this point, the interviewer read aloud the elements of both lists to the user and asked him or her for a justification. The answer had to be a specific explanation of the exact characteristics that made the mentioned elements strengths or weaknesses of the product [29]. Then, the user was asked for a solution to the problems (for negative elements) or improvements (for positive elements).

The users then completed a survey, which was composed of a simulator sickness questionnaire, usability aspects of warnings, and satisfaction questionnaires. The users named the warnings they remembered and relevant aspects and indicated how much they liked each warning variable on a 5-point Likert scale from 1 (very bad) to 5 (great). We thanked the users for participating in the study.

## 4. Results

Regarding the potentially influential users’ characteristics, our sample presented the following characteristics:Situational awareness coefficient of 0.89 for work situational awareness (reliability, which is acceptable if Cronbach’s alpha = 0.86) [85];Safety behavior—According to the mean scores, the respondents´ agreement/disagreement levels ranged from strongly disagree (mean = 1.75) to strongly agree (mean = 4.13);Safety experience—In terms of safety experience and, consequently, greater propensity to attention to prevent risk [86], 90% of the users had suffered or knew someone who had suffered a labor accident.

From the results obtained during immersion, it was possible to verify that the Game for Safety was engaging. By observing the video recordings, we found that warnings that were more perceptible were those that informed about hazards, as these were the first place that users looked. The signal with text was perceived by users, but they only stopped to read the dynamic signals (with LEDs). The signals on the floor were not perceived by the users; they claimed that this was because they were more concerned about the risks. These presented a high perception level. Figure 7 shows an example of a video recording extracted by navigation data.

With regard to simulation sickness, the data showed a low rate of discomfort, as shown in Figure 8. One possible cause of this distress could be if the experiment was conducted just after the user had eaten.

The evaluation of smart signals was discarded because of its low perceived ratings, caused by the elevated cognitive task due to its location. With regard to the static warnings scores, 80% of users considered them clear and 90% thought they were pleasant. The dynamic (LED) scores were 60% clear and 60% pleasant.

The results obtained in the BLA were classified according to the method with the following premises:Common positive elements (positive elements cited by more than one user—CPx);Particular positive elements (those mentioned by only one user—PPx);Common negative elements (negative elements cited by more than one user—CNx);Particular negative elements (items mentioned by only one user—PNx).

According to their positive or negative evaluations, the common elements mentioned at higher rates (except the CP average score 10.0) were the most important aspects to improve, use, adjust or modify. Table 3 shows the highlighted aspects identified by the users and their respective classifications. Additionally, a mention index was obtained to determine the elements that were perceived by the users (a high mention index indicates that the element was perceived more by users). The combination of these elements determined the most relevant items obtained from the BLA.

As shown in Table 3, the higher consensus on the positive aspects identified with an MI of 60% referred to the assessment of decision making. The second most rated characteristic was learning without risk (50% MI). This fact could prove the usefulness of Game for Safety to evaluate causes and consequences. Other positive issues cited with rates above 30% were realism, usability, time for the task, and the presence of warnings (all 40% MI). This last one deserves attention because it is related to issues that also appeared in particular comments with the highest average (LED warnings and warning pictograms). This fact can verify the importance of and attention to warnings in decision making. It is important to note that, for positive elements, negative scores tend to apply to elements that are perceived as bonus features that work badly [32]. For example, in the case of realism (Av. 4.75), the users perceived it to be positive as a concept, but their final experience was not pleasant.

For the negative issues, there was no MI above 40%. The most frequently mentioned issue was the presence of distractors. The users stated, for example, that the presence of hourglasses (extra bonus) aroused the desire to trigger them, although it was not a mandatory action. Additionally, in real accident situations, there is no extra time awarded for safety.

One interesting point is the fact of both realism and virtual environment were positively considered by women, while male participants rated them mostly with negative scores. This difference is aligned with previously findings in research about gender difference behaviors in virtual environments [87,88]. We consider this an important aspect which deserves further exploration in a quantitative study

The improvements and suggestions by the users were classified as common solutions (CS) when they were mentioned by more than one user and as particular solutions (PP) when they were cited by just one user, as shown in Table 4. It is interesting to note in this phase that the same suggestion was presented by more than one user as a solution or improvement for more than one item.

The three most cited suggestions were to adjust interactions to the user’s profile, reduce elements, and improve the layout. The users commented that the character’s locomotion should change according to the user’s profile by enabling or disabling the lateral displacement of the character according to how often the user utilized the technology. The suggestion of more information was related to sickness (inform users not to eat just before the experiment).

## 5. Conclusions and Future Work

The main contribution of this paper is the evaluation of variables to improve the design of serious virtual games to analyze the behavior of users in a risk environment as well as variable warnings to evaluate their effect on users’ interaction. We assessed the experience by measuring both subjective and behavioral measures. With regard to the results obtained from the video recording originated from navigation data, physical reaction, behavioral evidence, and questionnaire ratings, we addressed the following question:RQ1: Are there differences in the perception, behavior and usability of users for different variable signs as a function of the level of danger?

The outcomes of the survey showed a higher-level perception of warnings when they were endowed with dynamic technology variables. In this sense, we found that high percentages of participants in a gamified and stressful condition directed their vision first to a signal with an intermittent LED variable. In a textual poster, the LED variables were related to higher perception and reaction time. However, neither the color nor the disposition generated substantial effects on users.

Gravity and eminence did not produce distinct reactions in perception, behavior or decision making. All participants’ behavior remained constant for different types of hazards (solved or escaped).

Considering the data obtained in the study and our research question, we can provide the following insights.

RQ2: What elements could improve the design of a virtual environment for the analysis of user behavior in a gamified risk context?

In this sense, we identified two main elements: time and stress.

It has been shown that the pressure of finishing the principal task (crossing the maze) in less time in addition to the personal tension of addressing risks constitutes conflicting objectives.

According to the BLA evaluation, stress had three principal causes: repetitive sensation, time pressure and excess elements. A long maze with a similar layout provokes the feeling of constantly being in the same place. The disposition of various elements works as a strong distractor that deviates users. In this way, the simple reason of having an extra time bonus element added by the gamification surrounding encourages individuals to achieve the goal of earning time even if they do not need more time.

Therefore, we can say that the environment layout plays an important role in virtual environments. It is very important to consider the selection of the elements and their locations to motivate, distract or focus the user.

By crossing the three phase dates and considering our third research question, we can conclude the following:RQ3: Which components of an immersive serious risk game can impact the user?

Our results suggest that the possibility of evaluating cause and effect constitutes a powerful argument for the use of an immersive virtual environment. Additionally, the engagement produced by gamification stimulates training tools.

We also found that motivators in a virtual environment consist of better adjusting objectives to user profiles rather than enhancing elements. This happens, for example, because the system requirements of a person with game familiarity differ from those of a person without game familiarity. Game mechanics and the virtual environment therefore provide a level of engagement and interactivity that makes it a promising tool in any phase of the design process.

In relation to warnings, the present study demonstrates that technological advances provide sensors capable of increasing warnings. Therefore, the mode in which the signal is presented influences users’ perceptions. Sensors aimed at providing dynamic warnings are a good solution to improve the efficiency of signals. However, some changes should be made to evaluate smart signals. Based on the results of this study, we will investigate the range of possibilities of smart variables and their effects (e.g., changing the horizontal form at the floor, personalizing warnings, and implementing audio cues).

As a next step for this study, we will work to implement Game for Safety in a quantitative study. In this way, we will be able to conduct an objective analysis of the influences of warning variables to reduce accidents. We also want to explore the influence of gender in the behaviours.

## Figures and Tables

**Figure 1 sensors-21-04663-f001:**
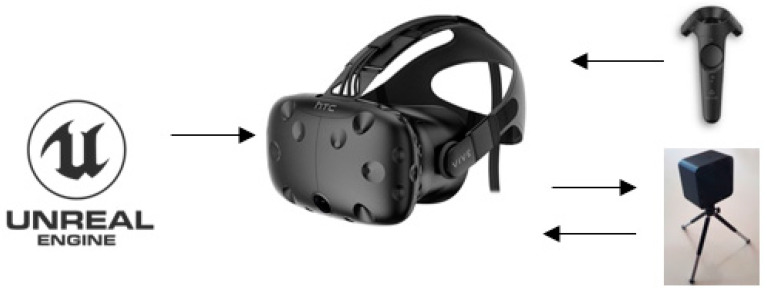
Scheme of immersive virtual reality components.

**Figure 2 sensors-21-04663-f002:**
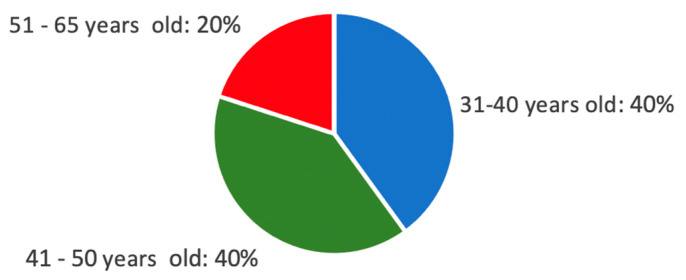
Percentage of age range.

**Figure 3 sensors-21-04663-f003:**
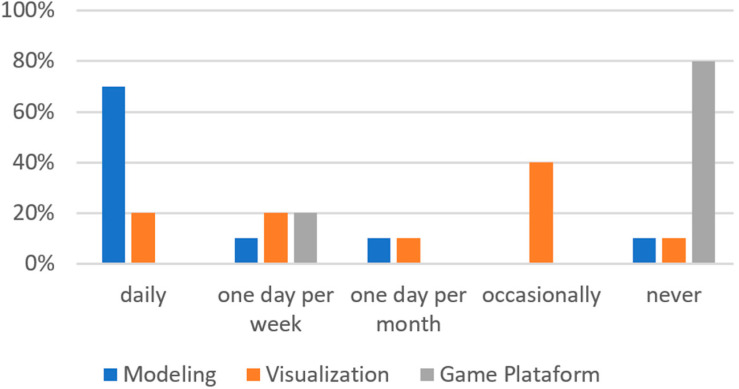
Use frequency of 3D platform.

**Figure 4 sensors-21-04663-f004:**
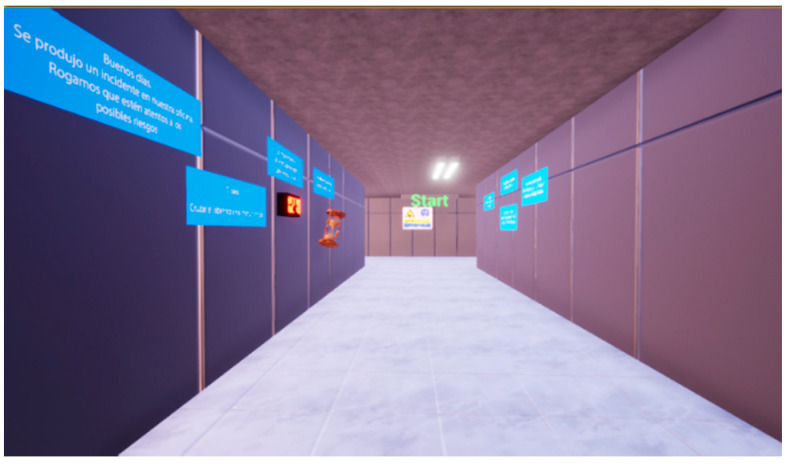
Pre-experiment training room.

**Figure 5 sensors-21-04663-f005:**
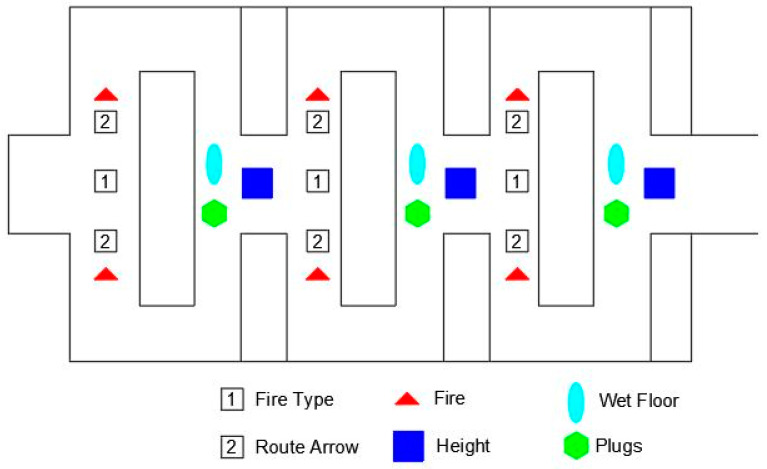
Hazard distribution, route and educational signs in the principal maze.

**Figure 6 sensors-21-04663-f006:**
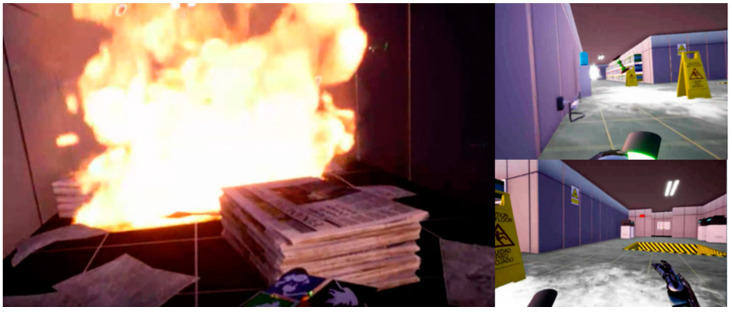
Example of hazards.

**Figure 7 sensors-21-04663-f007:**
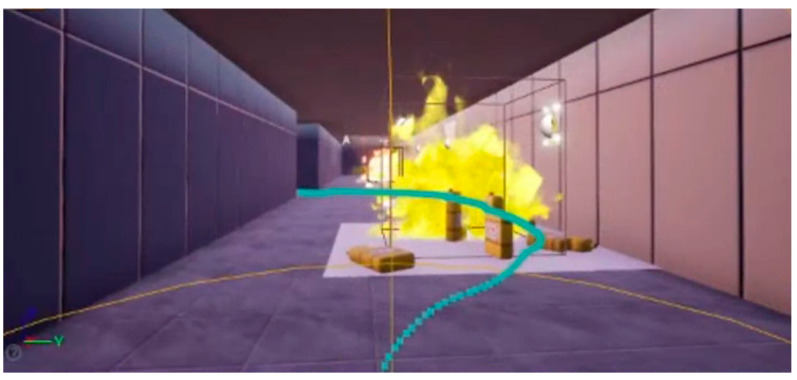
Screenshot of a video recording showing user’s path.

**Figure 8 sensors-21-04663-f008:**
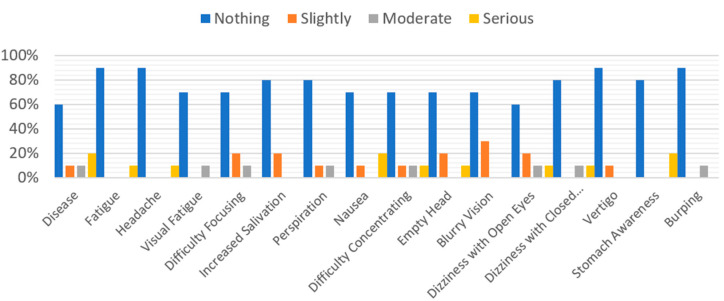
Simulator Sickness.

**Table 1 sensors-21-04663-t001:** Summary of the metrics.

Dependent Measures	Stage	Research Tools
Behavioral Intentions	Pretest	Questionnaires
Perception Times; Response; Reaction Time; Behavior; Compliance; Accuracy; Selective and Divided Attention	During the game	Navigation data, physical reactions, behavioral evidence
Memory; Ratings of Perceived Appropriateness; Qualitative Satisfaction	Post-experiment	Questionnaire ratings; Bipolar laddering

**Table 2 sensors-21-04663-t002:** Signals used in Game for Safety.

Static Signal	Dynamic Variable	Smart Variable
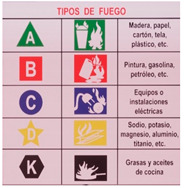	Intermittent LEDs at the extremes of the poster. Intermittent LEDs at the extremes of each cell information	------------------
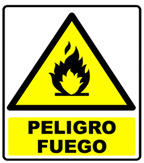	Intermittent white color LEDs Intermittent yellow color LEDs Intermittent red color LEDs	------------------
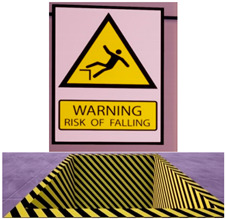	Intermittent white color LEDs at the posters	Poster and warning band appear when user advances until the danger
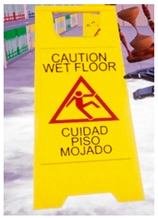	Intermittent LED lights around text Intermittent LED lights around pictograms	------------------
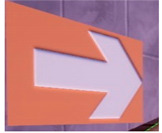	Intermittent LED lights on the arrow	Signal appears when user advances until the danger

**Table 3 sensors-21-04663-t003:** BLA positive common (PCx), negative common (NCx), positive particular (PPx), and negative particular (NPx) elements of the game and sensor warning variables.

Item ID	Description	Av. Score (Av)	Mention Index (MI) %
PC1	Learning without risk	8.40	50
PC2	Assessment of decision making	9.70	60
PC3	Presence	6.34	30
PC4	Realism	4.75	40
PC5	Virtual environment	8.50	20
PC6	Usability	9.75	40
PC7	Time x Task	8.50	40
PC8	Engagement	9.00	20
PC9	Warnings	9.75	40
PC10	LED warning	10.00	20
PP1	Warning pictograms	10.00	10
PP2	Game	9.00	10
PP3	Clarity	5.00	10
PP4	No aggressive/Pleasant experience	6.00	10
NC1	Get used to technology	6.00	20
NC2	Distractors	3.75	40
NC3	Fire extinguisher command	2.50	20
NC4	Sickness	4.67	30
NC5	Fire extinguisher usability	5.00	20
NC6	Repetitive	2.00	20
NC7	Realism	5.00	20
NC8	Bonus time	2.50	20
NC9	Graphic environment	2.50	20
NP1	Adaptability	4.00	10
NP2	Time information	4.00	10
NP3	Warnings with texts	6.00	10
NP4	Water	6.00	10
NP5	Duration	6.00	10
NP6	Environmental Lay-out	4.00	10

**Table 4 sensors-21-04663-t004:** Common and particular solutions identified for negative and positive elements.

Item ID	Description	Mention Index (MI) %
CS1	Longer experience	20
CS2	Adjust interaction to user profile	40
CS3	More initial information	20
CS4	Change time display	20
CS5	Reduce elements	40
CS6	Improve layout by area (colors, elements, events)	30
CS7	Less text on warnings	20
PS1	Limit time for fire extinguisher	10
PS2	Another level	10
PS3	Improve speed	10
PS4	More difficult to access fire extinguisher	10
PS5	More severe negative effect	10
PS6	More stressful elements	10
PS8	Change the water effect	10

## Data Availability

The data of the study can be available at this link: https://www.dropbox.com/s/pqv7uq12tl5ycax/Sensors%20.xlsx?dl=0 (accessed on 5 June 2021).

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
