# Peer review of "Gamification and Hazard Communication in Virtual Reality: A Qualitative Study"

_sensors, 2021, doi:10.3390/s21144663_

Round 1

Reviewer 1 Report

The paper il well presented. I suggest, since the journal the authors have applied is Sensors, to explain better the methodology used to create the virtual environment and a deepen explianation of the sensors used

Author Response

Dear Reviewer,

thank you so much for your time and efforts to review and improve our paper. Please see attached the detailed comments to your suggestions implemented in the new version of the manuscript.

Best Regards.,

Reviewer 2 Report

This research is timely and interesting. The authors carefully and thoroughly investigated the literature.

Regarding the format of citations, there are some inaccuracies in the introductory section (mostly). For example, the references are cited in the article as [x],[y],[z], while it could be [x,y,z] or [x-z]. This is inconsistent as well, because the citations are well-formatted later in the article.

It should be noted that gamification means that using elements lent from games outside of the field of games. My question is: what makes the authors' application a (serious) "game"? This is not understandable from the article. When the player character moves forward by pressing the forward button, it does not automatically mean that an application is a game. What makes this application a game?

Also, previous studies show that women react to VR or do tasks differently in VR than men. As there is data of both men and women, it would be interesting to see whether there are differences between their reactions, playstyles, etc.

English of the authors is good, although there are some mistakes. These are mostly typos, and few sentences are "weird". Here are some examples:

There are extra spaces in the legends of Fig. 2
"Game Platafom", "ocasionaly" (Fig. 3)

"The kinematics recorded then began." (line 330)

The latter sentence has no meaning (it can be guessed, however). There are also some sentences where are too many definite articles ("the"). Please, correct these (and other) spelling mistakes in the article.

The format of the references are mostly correct, however e.g. the family names of a few people in reference no. 7 are not written fully, only as abbreviations.

Author Response

(The authors gave the same response as above.)

Reviewer 3 Report

I liked your paper very much and it seems interesting to me. Especially because it is the preliminary stage of a quantitative analysis, I am very curious about the results that will then be collected. 
If possible, please correct the spelling mistakes.
Otherwise, the paper should be published.

Author Response

(The authors gave the same response as above.)
